# A Real-Time Deep Machine Learning Approach for Sudden Tool Failure Prediction and Prevention in Machining Processes

**DOI:** 10.3390/s23083894

**Published:** 2023-04-11

**Authors:** Mahmoud Hassan, Ahmad Sadek, Helmi Attia

**Affiliations:** 1Hybrid Manufacturing, Aerospace Manufacturing Technologies Center (AMTC), National Research Council Canada, Ottawa, ON K1A 0R6, Canada; 2Department of Mechanical Engineering, McGill University, Montreal, QC H3A 0C3, Canada

**Keywords:** cutting, condition monitoring, prefailure prediction

## Abstract

Tool Condition Monitoring systems are essential to achieve the desired industrial competitive advantage in terms of reducing costs, increasing productivity, improving quality, and preventing machined part damage. A sudden tool failure is analytically unpredictable due to the high dynamics of the machining process in the industrial environment. Therefore, a system for detecting and preventing sudden tool failures was developed for real-time implementation. A discrete wavelet transform lifting scheme (DWT) was developed to extract a time-frequency representation of the AE_rms_ signals. A long short-term memory (LSTM) autoencoder was developed to compress and reconstruct the DWT features. The variations between the reconstructed and the original DWT representations due to the induced acoustic emissions (AE) waves during unstable crack propagation were used as a prefailure indicator. Based on the statistics of the LSTM autoencoder training process, a threshold was defined to detect tool prefailure regardless of the cutting conditions. Experimental validation results demonstrated the ability of the developed approach to accurately predict sudden tool failures before they occur and allow enough time to take corrective action to protect the machined part. The developed approach overcomes the limitations of the prefailure detection approach available in the literature in terms of defining a threshold function and sensitivity to chip adhesion-separation phenomenon during the machining of hard-to-cut materials.

## 1. Introduction

In the machining of difficult-to-cut materials, the generated high mechanical and thermal loads on the cutting edge can lead to different mechanisms of tool failure in terms of tool wear, chipping, or breakage [1]. Each of these modes of tool failure has different mechanisms, types, and consequences depending on the tool load and the cutting process. The development of these failure mechanisms is also affected by the cutting conditions and the tool-workpiece engagement. Consequently, this leads to different changes in the tool edge geometry. As a result, different machined part quality attributes are produced, such as deviations in dimensional accuracy and surface integrity. To protect the machined part, manufacturers use a conservative tool change policy, where the cutting tool life is not fully utilized [2]. This increases the process cost and downtime. Therefore, it is essential to continuously monitor the tool condition in real-time to increase process productivity and reduce process costs while maintaining the machine part quality. Due to the complexity of the machining process, empirical models such as Taylor’s tool life equation may not be accurate enough to predict tool life [3]. Further, due to the complexities involved and the multiple wear mechanisms seen in machining, there is no unique solution for describing the complete milling process [4]. An effective real-time sensor-based tool condition monitoring (TCM) system can put a cutting tool under surveillance to safeguard the workpiece from damage by dealing with the uncertainty of analytical tool life prediction.

To detect tool chipping or breakage in milling processes, the extracted features from sensor feedback signals need to fulfill the following requirements: (a) must represent tool breakage under variable cutting conditions and different workpiece and tool materials, and (b) must be uniquely distinguishable to avoid interference with other process irregularities, e.g., tool-workpiece interactions, material inclusions or complex geometric features. An abrupt tool geometry change due to brittle fracture or breakage can alter the characteristics of the acquired signals. The latter includes a sudden burst in the acquired forces, power, or vibrations due to a sudden change in the cutting edge, and subsequently, the interaction between the tool and the workpiece. In addition, the generation of new surfaces associated with the separation of major fragments of the tool material at fracture releases high elastic waves. These waves can be observed as high energy in the acoustic emission (AE) signal. In addition to the ability of the AE sensor to capture abnormalities during the cutting process, it provides the flexibility for being mounted on any of the components of the machining system without major disturbances to various machining activities [5]. In addition, the frequency level of the AE signals produced from cutting processes has been found to be separable from audible noise [6]. In general, for TCM, an AE sensor mounted on a spindle or cutting tool showed higher reliability than when mounted on the workpiece [7,8,9]. Most of the efforts in modeling acoustic emission in manufacturing processes are built on the same model [10,11]. This model is based on the dependency of AE energy on material properties, such as flow stress, the volume of material undergoing deformation, and the strain rate. However, the influence of feed and depth of cut variations are not accurately understood. Moreover, it was found, in practice, that AE signals were corrupted with white noise generated from sources such as electron movements during signal transmission [12]. Hence, more attention is required for conditioning AE signals.

In contrast to tool wear, which is a progressive deterioration phenomenon, tool chipping and breakage are sudden tool failures because they are stochastic phenomena that occur in milliseconds. Unstable crack propagation preceding the separation of fragments of the cutting tool edge is considered the prefailure phase for sudden tool failure [13]. The large body of research conducted to develop and automate TCM systems has focused on detecting the failure event onset or its following effects [2,14,15]. By which time, the workpiece surface integrity and part quality may be impaired.

The detection of the prefailure phase of sudden tool failures has been investigated in previous work by the authors. A method was devised to induce cyclic load on the cutting tool tip to study the effect of unstable crack propagation, which preceded tool chipping, on indirect sensing methods such as cutting forces, vibrations, power, and AE signals [13]. Forces and vibrations were shown to be sensitive to the onset of tool chipping, whereas power was shown to be insensitive to the event. The acoustic emission AE signals were shown to be sensitive to the elastic waves generated in association with the generation of new surfaces during the unstable crack propagation phase.

In intermittent cutting processes, identifying the elastic waves associated with unstable crack propagation in raw AE signals is challenging due to the bursts coming from other process sources, e.g., impacts at inlet and exit, chip formations, rubbing between the tool and workpiece, and plastic deformation in the primary, secondary, and tertiary shear zones [16]. The unstable crack propagation effect on the AE signals is characterized by infinitesimal time spans of high energy/frequency bursts. In addition, the AE signals generated are nonlinear, non-stationary, and contaminated by bursts coming from the force variation in the intermittent cutting processes. To emphasize the crack propagation effect on the AE signals and to depress the steady state cutting effect, a novel approach based on the Hilbert-Huang Transform (HHT) and the Teager-Kaiser Energy Operator (TKEO) was developed by the authors to detect the prefailure evolution phase [17]. This TKEO_HHT approach was integrated with a 2-way communication module to predict tool failure in real-time and to stop the process before tool chipping and part damage. This method depends on the relative changes in the instantaneous amplitude and frequency of two sequential AE segments to detect tool prefailure. The TKEO_HHT was applied in real-time monitoring and control of intermittent turning, as well as conventional and high-speed milling operations [17,18]. The approach showed high prediction accuracy under a wide range of cutting speeds, feed rates, and axial and radial depths of cut during the machining of aluminum and steel workpieces. However, a main limitation of this approach was the experimental learning process needed to define the chipping threshold. This requires running controlled tests to generate tool chipping and correlate its size to the processed signals. Such a requirement limits its application in industrial facilities. Additionally, during the machining of difficult-to-cut materials, such as Ti-alloys, the high thermal loads lead to chip adhesion on the tool cutting edge. Once this chip is separated, this releases elastic waves due to the generation of new surfaces. The repetitive adhesion-separation phenomenon of the machining chips on the cutting edge is similar to the effect of unstable crack propagation on the AE signals, which causes false alarms and decreases the TKEO_HHT prediction accuracy.

Transient elastic waves generated from unstable crack propagation cause an instantaneous deviation in the acquired AE signals. Such deviation can be detected using anomaly detection approaches. These approaches identify data patterns that deviate from expected behavior, leading to the timely identification of underlying problems that can lead to costly consequences. The development of an anomaly detection approach mainly depends on the nature of the anomaly and the type of input data. For sensory data, anomalies can be categorized by faults such as drift, noise, constant, or spike [19]. The latter includes the effect of the unstable crack propagation on AE signals as shown in Section 2. Typically, an anomaly detection approach combines feature extraction techniques, such as principal component analysis and Euclidean distance, with machine learning approaches, such as classification, clustering, and regression models [20]. Several conventional and deep machine learning (ML)-based approaches have been developed in the literature to detect spike anomalies in sensory data. However, deep ML approaches have shown better capability in dealing with anomaly detection challenges in terms of input data format and noises and anomaly data scarcity and complexity [21]. Luo et al. utilized autoencoders to develop a distributed anomaly detection technique for spike detection in 1-D data [22]. Chen et al. proposed Spectral and Time Autoencoder Learning to detect drift, shift, and spike in time-series sensory data [23]. For model optimization to enable real-time implementation, Ding et al. developed a diagnosis network based on weight-sharing multiscale convolutions to extract multi-time scale features while minimizing the computational time [24]. In this work, an anomaly detection approach based on deep machine learning and wavelet analysis algorithms is introduced. The approach masks the effect of the chip adhesion-separation phenomenon on the AE signals for accurate detection of the prefailure phase during the machining of difficult-to-cut materials. It also overcomes the learning limitations of the TKEO_HHT approach, which facilitates the implementation of the prefailure detection system in an industrial environment.

This article is organized as follows: Section 2 discusses the sources and characteristics of the AE signals in intermittent cutting operations as well as the relation between the process parameters and the generated AE signals during sudden tool failure. Section 3 presents the development of the elements of the proposed tool prefailure detection approach. It describes the contribution of each processing step in accentuating the elastic waves associated with unstable crack propagation in AE signals and the optimization required to implement the approach in real-time. Section 4 lists the experimental setup, machine instrumentation, and test matrix for developing and validating the proposed approach. Section 5 illustrates and evaluates the results of applying the proposed approach for prefailure detection. The approach to developing and validating results is examined and benchmarked in comparison with a state-of-the-art prefailure detection approach available in literature. The time provided by the approach to take corrective action in order to safeguard the machined part is also discussed. The significant reduction in the calibration and training effort is demonstrated. Finally, the development approach methodology, validating results, and industrial impact are concluded in Section 6.

## 2. Acoustic Emissions in Intermittent Cutting Operations

Acoustic emissions derived from material deflection, chip breakdown, and pulse shock loading are produced during cutting operations. The cutting state is revealed by both the continuous and transient AE signals [12]. Continuous AE signals are made up of overlapping transient signals. These are associated with shearing in the primary zone, formation, and collisions of chips, and rubbing between cutting tools, workpieces, and formed chips. Transient AE signals are generated when a pulse shock loading occurs, such as chip breakage, entry/exit of each individual tooth to the workpiece, and tool vibrations and damage. During intermittent complex cutting processes, identifying the prefailure phase is challenging due to the bursts coming from other process sources [16]. The indeterminateness of some of these events in addition to the stochastic nature of the unstable crack propagation and the generated AE waves cause the non-stationary and non-linear nature of the AE signals.

The root mean square values of the acoustic emission signals (*AE_rms_*) have the same time-domain sensitivity to the tool prefailure phase as the raw AE signals [17,18]. However, the *AE_rms_* signals have the advantage of reducing the required sampling rate, which minimizes data storage. Therefore, the *AE_rms_* signals were utilized for tool prefailure detection. To describe the generated *AE_rms_* signals during cutting, a quantitative model of the *AE_rms_* peak voltage in machining using carbide inserts was successfully developed [25]. This model was developed in order to understand the AE signal response to the fracture of carbide inserts during intermittent cutting. The model describes the *AE_rms_* voltage as a function of the cutting tool material properties, wave propagation properties, crack propagation, and cutting forces as follows:(1)$$AE_{rms}=K_{1}\sqrt{\frac{E}{{\left(1+v\right)}^{2}\cdot \left(1-v^{2}\right)}}\sqrt{\frac{\omega^{2}\cdot \delta^{2}}{\left(k^{2}+\omega^{2}\right)}}F_{r}\sqrt{\alpha a^{\beta }\cdot \Delta a\cdot \Delta {A_{c}}^{2}}$$
where *K*_1_ is a constant, *E* and v are the modulus of elasticity and Poisson ratio of the tool material, respectively; k, ω, and *δ* are the crack AE wave decay constant, the frequency of decaying and the stress propagation factor, respectively. The symbol *F_r_* stands for the resultant cutting force at tool fracture, *α* and *β* are constants related to tool geometry, and *a* and *A_c_* are the crack length and area, respectively. This equation can be simplified, for the same tool material, after assuming that $\Delta A_{c}$ is a linear function of Δa, as follows:(2)$$AE_{rms}\approx CF_{r}{\left(\Delta A_{c}\right)}^{1.5}$$
where *C* is a material and geometry-dependent constant. Equations (1) and (2) show the nonlinear relationship between the *AE_rms_* signal and cutting forces. The *AE_rms_* signal variation through the course of the intermittent cutting operation, for the same insert, depends mainly on the resultant force variation and the area of the newly generated crack surfaces.

## 3. Anomaly Detection Approach for Sudden Tool Failure Prediction

This work proposes an anomaly detection approach that integrates time-frequency signal analysis with deep machine learning approaches to define unstable crack propagation in cutting tools. It employs a discrete wavelet transform algorithm (DWT) to extract representative features of the AE waves generated during normal operating conditions. Sequentially, these features are fed to a long short-term memory artificial neural network autoencoder (LSTM) to predict tool prefailure. This DWT-LSTM autoencoder allows accurate detection of the prefailure stage and overcomes the limitations of the available TKEO-HHT approach in terms of learning effort and false alarms in hard-to-cut materials.

### 3.1. Discrete Wavelet Transform

A transient *AE_rms_* wave is typically a nonlinear signal that exhibits a shape that reflects impacting and exponential decay properties [2]. A wavelet transform decomposes this signal into a family of wavelets by creating a set of functions. Each wavelet represents a specific frequency band and creates time-localized frequency components. Therefore, transient AE signals with discontinuous and sharp changes, which are temporally localized, can be captured. This facilitates extracting representative features.

In this work, a Daubechies orthogonal wavelet lifting scheme (DWT) was adopted [26]. This DWT signal processing approach provides multi-resolution signal details. In the first level of this analysis, acquired *AE_rms_* signals are divided into two components based on the signal frequency using a scaling function, a wavelet function, and filter banks. The scaling function provides a coarse representation of the signal, while the output of the wavelet function defines the signal details. A hierarchical representation of the *AE_rms_* signal is developed by recursively processing the low pass output of the filter bank as an input signal. The cascading process over this hierarchical representation leads to a sum of coefficients at different signal resolutions and a residue. These coefficients can be used to reconstruct the signals. The scaling function decreases the signal size at each step, which reduces the computational time. Daubechies’ DWT was selected due to its capability to capture sharp and irregular changes in the signals [27]. To emphasize transient events in AE signals, a Daubechies wavelet db2 was selected, which offers relatively smaller support to separate the features of interest.

### 3.2. LSTM Autoencoder Neural Network

An autoencoder is a machine-learning approach that leverages neural networks for efficient data representation. It consists of an encoder and a decoder. The encoder decomposes the input data into a compressed representation of itself by extracting features with low dimensionality. The decoder then uses this representation to reconstruct the input data with its original dimension. Through the training stage, the autoencoder learns to regenerate the input signals with high accuracy. This accuracy decreases when the autoencoder tries to regenerate an input signal that contains outliers. The decrease in accuracy is used as an indicator of abnormality.

Different architectures have been implemented in developing autoencoders, including Conventional, feedforward, and long short-term memory (LSTM) networks [28,29,30,31,32]. In this work, the latter was adopted due to its high flexibility and adaptability [33]. LSTM networks define the compressed signal features by storing representations of the input signal pattern over time steps [34]. This enhances the LSTM performance to capture the non-linear dynamics in the input signals (i.e., DWT outputs). The LSTM structure consists of a cell state and an output state that are updated during the learning process. The prediction from earlier time steps is kept in the cell state, whereas the layer output for a particular time step is kept in the output state. By examining the input signal in advance, the unidirectional LSTM (ULSTM) learns the state of each of its cells. In order to update the cell state of the input signal based on the past and future output states simultaneously, bidirectional LSTM (BiLSTM) adds another backward scanning to the input signal [35]. Despite the fact that such an activity might over-restrain the generated model, BiLSTM was selected to increase the model sensitivity to the training data, and hence, accentuate the abnormal events’ effect on the prediction accuracy.

A deep recurrent neural network consisting of five BiLSTM layers was developed. The input data was compressed using a falling number of hidden units as it was encoded from layer one to layer three. Then, a mirrored order of the hidden unit numbers used in layers one to three is employed to build the reconstructed version of the input data from layers three to five. An adaptive moment estimation (Adam) optimizer alongside with mini-batch size and a maximum number of epochs of 32 and 40, respectively, were implemented in this study to train the LSTM autoencoder.

### 3.3. DWT-LSTM Implementation

Figure 1 shows the flowchart of the proposed DWT-LSTM autoencoder approach. The machining AE-generated signals typically fall into the range of 100–600 kHz, whereas the tool chipping or fracture causes high-powered oscillations in the frequency range from 300 kHz to 1.0 MHz [6]. Hence, acoustic emission signals were amplified and filtered using a bandpass filter of a 100 to 1000 kHz bandwidth. This filtering range selection represents the typical range of AE signals generated from tool chipping and fracture while eliminating those coming from mechanical vibrations and audible noise, which are below 20 kHz. The signal root mean square *AE_rms_* is then calculated and segmented per workpiece revolution. The developed DWT lifting scheme is applied to extract the time-frequency information of the *AE_rms_* signals and feeds it to the LSTM autoencoder. The statistics of the training signals reconstruction error were used to define a prefailure threshold for real-time implementation. The mean square error (MSE) between the LSTM-autoencoder training data and predictions was calculated for each training segment. Following a Gaussian distribution, a prefailure detection threshold was selected to be three times the MSE standard deviation.

## 4. Experimental Setup and Test Matrix

This research mainly focused on cracks due to mechanical loads only while avoiding the occurrence of tool wear and heat build-up. Therefore, a method was devised to induce a cyclic impact load on the cutting tool tip in an intermittent turning operation, as shown in Figure 2. This represented the loading conditions in milling as well. This type of test allowed testing the developed approach capability to capture the unstable crack propagation and tool edge chipping while ensuring minimal tool wear. To induce a cyclic impact load on the tooltip, a Ti-6Al-4V workpiece was used in the shape of a plate. The plate thickness to width ratio was selected to allow air cooling during 85% of the cutting revolution. The plate-holding seats were designed to guarantee workpiece balance during the cutting operations. Dry turning operations were carried out using a SECO DCLNR2525X12JETI tool holder and SECO CNMG120408-MF4 TS2000 carbide inserts. Tests were conducted on a 6-axis Boehringer-NG200 CNC turning center. This machine tool has a maximum spindle power and rotational speed of 36 kW and 4000 rpm, respectively. Workpieces were pre-shaped as seen in this figure to minimize the transient stage of tool entry. FASTCAM high-speed camera (HSC) type UX100-800K-M was used to record the chipping events in real-time during cutting. This HSC provides 1280 × 1024 pixels resolution with a selectable region of interest. It has a maximum frame sampling rate of 800 kfps and can be triggered to start recording using selectable +/− TTL 5 V and switch closure with a response time of 0.1 μs. These characteristics provided the ability to evaluate the tool condition in synchronization with the acquired signals for analysis purposes. The synchronized imaging of this HSC was used to detect the chipping events during the cutting processes and to relate them to the acquired signals. The tool holder was mounted on a three-component KISTLER dynamometer type 9121 to measure the cutting forces with a measurement error of ±3%. The force signals were amplified using a KISTLER 5010 amplifier. A miniature triaxial PCB accelerometer type 356A71 was used to acquire vibrations during the cutting processes. It has a sensitivity and a measuring range of 1.02 pC/(m/s^2^) and 500 gpk, respectively. Acquired signals were conditioned and amplified using a PCB signal conditioner model 480C64. AE-generated signals were captured using a KISTLER Piezotron AE sensor type 8152B and conditioned using a KISTLER AE coupler model 5125C1. The sensor was mounted on the back of the cutting tool to be as close as possible to the cutting zone. The cutting force and vibration signals, as well as the timing of chipping initiation as recorded by the HSC imaging, were employed to define the onset of fracture, and compare it to the AE-based prefailure indicator.

In this work, four experiments were conducted with various speeds, feed rates, and cut depths. The cutting conditions and role of each test are shown in Table 1. Test 1 cutting passes were performed for the DWT-LSTM autoencoder training, utilizing a fixed depth of cut of 1 mm, a range of speeds between 35 and 75 m/min, and a range of feed rates between 50 and 100 mm/min. Three experiments were carried out to validate the developed approach. The cutting conditions of test 2 were selected within the training range. Tests 3 and 4 were carried out under unlearned cutting conditions in order to demonstrate the approach’s generalization capability. Both tests had a higher depth of cut and feed rate.

## 5. Results and Discussion

### 5.1. DWT-LSTM Processing Approach Capabilities

*AE_rms_* signals acquired during Test 1 were segmented per revolution and used for training the developed DWT-LSTM autoencoder. In total, 10 cutting passes were conducted using different combinations of the range of the cutting conditions of Test 1 shown in Table 1. To demonstrate the LSTM autoencoder accuracy in reconstructing the DWT features, Figure 3a,b shows a normalized training segment sample of the training *AE_rms_* and the outputs of the DWT and the LSTM autoencoder. The MSE for the illustrated segment was 1.3 × 10^−6^, whilst the average MSE of all training segments was 1.8 × 10^−6^. Similar reconstruction MSE was observed for AE signals acquired from Tests 2, 3, and 4, which were conducted using different cutting conditions. For example, Figure 3c,d illustrates the *AE_rms_* and the corresponding DWT and LSTM outputs for a normalized segment of Test 3 during normal cutting conditions, where a reconstruction MSE of 1.7 × 10^−6^ was achieved. This demonstrates the LSTM autoencoder’s capability to accurately reconstruct the DWT features regardless of the cutting conditions.

In Figure 4, Test 3 is illustrated as an example to demonstrate the capability of the DWT-LSTM approach to accurately detect the prefailure phase in the nonlinear *AE_rms_* signals. Figure 4a,d shows the raw force and *AE_rms_* signals, respectively, for 2.5 s. These signals were acquired during the prefailure phase and the chipping of the cutting tool. In this test, a chipping of 8.16 mm^2^, measured following the procedure described in [17], was observed on the cutting insert tip after 31.7 s of cutting, as shown in Figure 4c. The chipping event captured by the high-speed camera was confirmed by both the force and *AE_rms_* signals, represented by the high amplitude at fracture followed by a low amplitude in the sequential peak due to the reduced contact between the chipped tool tip and workpiece, as shown in Figure 4a,d. The vibration signals have shown similar responses during chipping as well. However, there was no sign of the prefailure phase in these signals. Starting at time *t* = 30.8 s, the DWT-LSTM approach has discriminated the effect of the unstable crack propagation during the prefailure phase using the error between the DWT features and their reconstruction by the LSTM autoencoder. This can be easily visualized by comparing the variation between the DWT and LSTM outputs in Figure 4e to Figure 3d. This reconstruction error is due to the high energy components induced by the new AE waves associated with the generation of new surfaces in the unstable crack propagation, which was not included in the DWT-LSTM autoencoder learning process. During this phase, the LSTM autoencoder MSE has increased by approximately 19 folds, compared to the MSE during machining using a sound tool before t = 30.8 s. The proposed approach is thus able to distinguish between the signals inherent in the cutting process, even in the presence of impact load conditions, and the non-stationary and non-linear signals of the prefailure phase, which is associated with the stress waves released by the new surfaces’ formation.

### 5.2. Online Implementation and Benchmarking of the DWT-LSTM Autoencoder

In this section, the DWT-LSTM autoencoder capability to detect tool prefailure phase in real-time is demonstrated and benchmarked with respect to the TKEO-HHT approach defined in [17]. The statistics of the MSE of the DWT-LSTM autoencoder were employed to define a threshold for prefailure detection as explained in Section 3. The MSE was calculated using the training data of Test 1 only, where no chipping event took place. This overcomes the limitation of the TKEO-HHT approach for defining a threshold value, where a chipping event needs to be captured during controlled cutting tests and correlated to the approach output. Additionally, the TKEO-HHT threshold is sensitive to the cutting conditions, as shown in Equation (2). Table 2 demonstrates processing time and the prefailure detection window available from the first prefailure indicator to the onset of chipping for cutting tests 2, 3, and 4 using DWT-LSTM and TKEO-HHT approaches. The DWT-LSTM technique has successfully forecasted the chipping events within a range from 540 to 830 ms before the chipping onset. The prefailure detection window for the TKEO-HHT method ranged from 600 to 970 ms. The two approaches showed a comparable prefailure detection window, with variations in window size between the two approaches and between different tests for the same approach. The variations in the time window sizes between the two approaches are related to their different prefailure detection concepts. The DWT-LSTM depends on the variation in the *AE_rms_* caused by the prefailure phase in comparison to signals acquired during normal cutting conditions, while the TKEO-HHT depends on the relative changes in the instantaneous amplitude and frequency between two sequential *AE_rms_* segments. The variations in the time window sizes within the same approach, however, are related to the crack size and propagation rate, which affects the strength of the produced AE signals during prefailure.

At a maximum processing time of 1.4 ms, the DWT-LSTM approach provided a shorter processing time compared to the TKEO-HHT approach. This is due to the HHT’s signal-sifting method, which necessitated a longer processing time of up to 3.8 ms [17]. However, both approaches provided enough time for real-time correction actions to protect the machined part in terms of feed hold command. This can be achieved through a 2-way communication controller with the CNC machine control to overwrite machine cutting parameters in real-time [18].

During Test 1 passes, the chip adhesion-separation phenomenon took place on several occasions. The effect of this phenomenon on the *AE_rms_* signals was included in the DWT-LSTM training as it is a data-driven process. This facilitates overcoming the false alarms caused by this phenomenon when the TKEO-HHT approach is applied. Figure 5 shows the *AE_rms_* signals of Test 4 and the corresponding DWT-LSTM and the TKEO-HHT prefailure detection indicators. In this test, tool chipping occurred after 15.8 s of cutting, as shown in Figure 5a. The DWT-LSTM and the TKEO-HHT approaches have successfully detected the prefailure phase 540 ms and 850 ms before fracture, respectively. However, the TKEO-HHT approach has shown a false alarm in terms of high prefailure indicator amplitude at *t* = 13.3 s, shown in Figure 5c. The imaging of this cutting process using the high-speed camera has shown that the chip adhesion-separation phenomenon took place during this period. Such misclassification is caused by the induced AE waves associated with the new surface generated during the chip separation. The DWT-LSTM approach has overcome this drawback, owing to the proposed learning approach, as shown in Figure 5b. This demonstrates the developed approach’s capability to accurately detect the prefailure phase in difficult-to-cut materials. Hence, the proposed approach responds to industrial needs in a real working environment. It facilitates full tool life utilization, while safeguarding the machined part and can be retrofitted to existing machine tools. This eliminates the costs accompanied by sudden cutting tool failure in terms of machine downtime and re-machining of defective parts or part scrap.

## 6. Conclusions

A sudden tool prefailure detection and prevention approach has been developed in this work. A discrete wavelet transform lifting scheme DWT was developed to extract a time-frequency representation of the *AE_rms_* signals. A Long short-term memory LSTM autoencoder was developed to compress and reconstruct the DWT features. The variation between the reconstructed and original DWT representation due to the induced AE waves during the unstable crack propagation was used as a prefailure indicator. A threshold to detect prefailure was defined based on the statistics of the LSTM autoencoder training process. The data-driven training of the DWT-LSTM approach has accounted for the AE variation due to the chip adhesion-separation phenomenon during the machining of hard-to-cut materials. The LSTM autoencoder has shown an average MSE of 1.8 × 10^−6^ for both learned and unlearned data, which proves the developed approach’s robustness in reconstructing the DWT of the AE signals regardless of the cutting conditions. Experimental validation tests have shown the DWT-LSTM approach’s capability to accurately predict the tool failure by up to 830 ms before it happens. This was achieved using only 1.4 ms of processing time, which provides enough time to take corrective actions by stopping the feed drive of the machine tool to safeguard the machined part. The approach was benchmarked to the available prefailure detection approach presented by the authors earlier in the literature and showed comparable prefailure detection time windows. However, at low computational and training costs, the DWT-LSTM approach overcomes the literature approach limitations in terms of defining the thresholding function and the sensitivity to the chip adhesion-separations phenomenon. This facilitates the implementation of the new proposed approach in industrial facilities. The proposed approach offers full tool life utilization while protecting machined parts and decreasing machining downtime and costs.

## Figures and Tables

**Figure 1 sensors-23-03894-f001:**
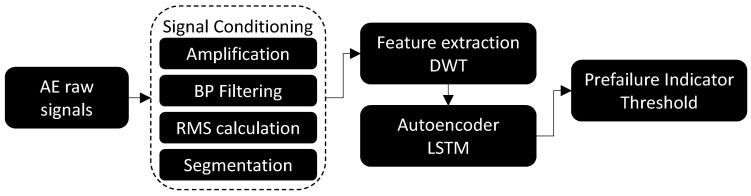
Anomaly detection approach flow chart.

**Figure 2 sensors-23-03894-f002:**
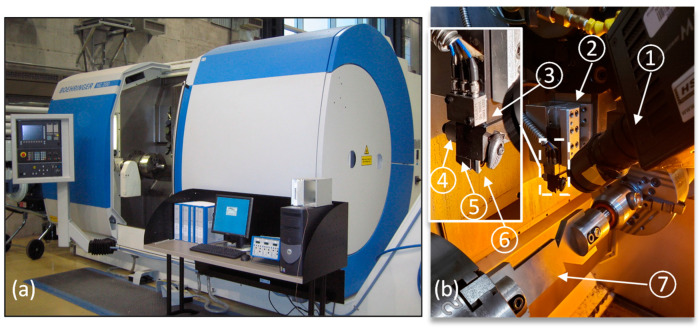
(**a**) Boehringer NG200 Turning Centre and (**b**) experimental setup. ① High-speed camera, ② Dynamometer, ③ Accelerometer, ④ AE sensor, ⑤ Cutting tool, ⑥ Cutting insert, and ⑦ workpiece.

**Figure 3 sensors-23-03894-f003:**
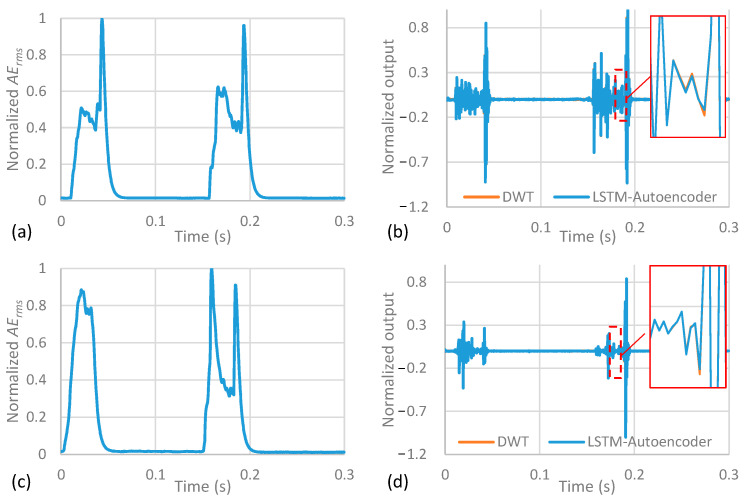
(**a**,**c**) Normalized *AE_rms_* signals and (**b**,**d**) the DWT output and its reconstructed representation from the LSTM Autoencoder of Tests number 1 and 3, respectively.

**Figure 4 sensors-23-03894-f004:**
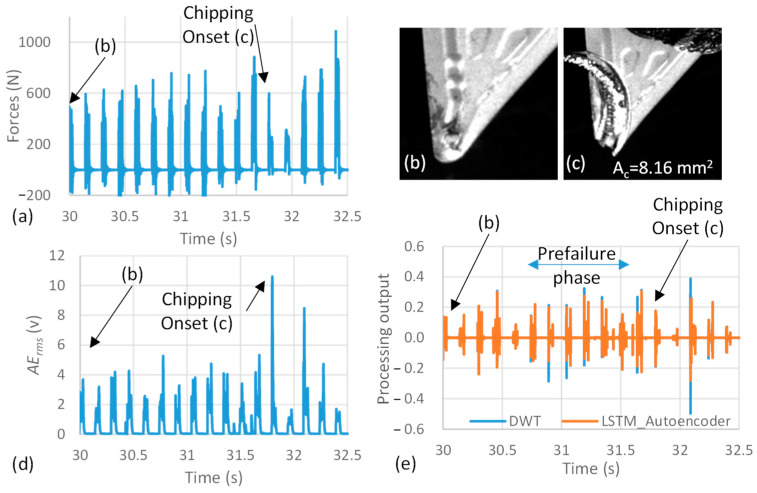
(**a**) Raw cutting force signals, (**b**,**c**) high-speed camera photos, (**d**,**e**) the raw and processed *AE_rms_* signal of Test 3 at tool failure.

**Figure 5 sensors-23-03894-f005:**
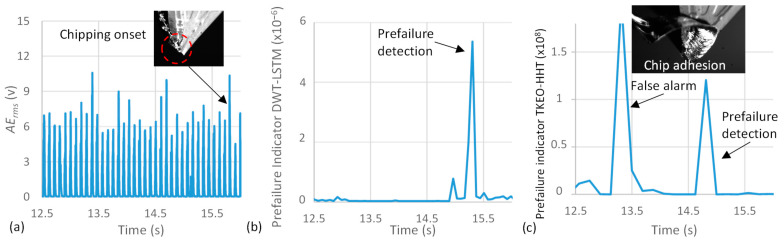
(**a**) Test 4 raw *AErms* signal, the corresponding (**b**) DWT-LSTM, and (**c**) TKEO-HHT prefailure indicators.

**Table 1 sensors-23-03894-t001:** Test matrix.

Role	Test Number	Speed (m/min)	Feed (mm/min)	Depth of Cut (mm)
Training	1	35–45	50–90	1
Validation	2	45	80	1
Validation	3	35	100	1.25
Validation	4	75	95	1.5

**Table 2 sensors-23-03894-t002:** Prefailure detection window and processing time of the DWT-LSTM and TKEO-HHT approaches.

	Prefailure Detection Window (ms)		Processing Time (ms)	
Test Number	DWT-LSTM	TKEO-HHT	DWT-LSTM	TKEO-HHT
2	830	600	1.3	1–3.8
3	660	970	1.1	1.3–4
4	540	850	1.4	1.4–3.4

## Data Availability

Data is unavailable due to confidentiality restrictions.

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
