# Peer review of "A Real-Time Deep Machine Learning Approach for Sudden Tool Failure Prediction and Prevention in Machining Processes"

_sensors, 2023, doi:10.3390/s23083894_

Round 1
Reviewer 1 Report
1. The results have to be added to the abstract.
2. What is the central question addressed by this research?
3. Research gaps and novelty are not presented in the manuscript.
4. Indicate the contribution of this paper to the manufacturing industry.
5. On what basis the authors selected the algorithms?
6. In Figure 3, DWT signals are not visible.
7. In Figure 4 b&c, the tool wear value can be indicated.
8. For classification accuracy, a confusion matrix needs to be added to the algorithms
9. A separate discussion section is mandatory for publication.
10. Proposed algorithm needs to be validated with the benchmarking dataset.
11. Conclusion needs to be strengthened with research findings and address the problem considered in the research.
Reviewer 2 Report
Dear Authors,
I found the article of a great interesting and clear.
I only suggest you some minor revisions:
1. Highlight better the novelty and the contributions of the paper in the introduction, not only referring to your previous work but also referring to the existing literature
2. Reduce the description of your previous approach to give more importance to the one proposed in this paper.
3. At the end of the introduction, describe the structure of the article (what each section includes) to facilitate the reading
4. Provide a more comprehensive explanation of the results in the conclusion
5. Please define the type of paper at the top of the first page (article, review, etc...)
Reviewer 3 Report
This paper introduces a real-time sudden tool failure detection method based on discrete wavelet transform (DWT) and long short-term memory (LSTM). This work seems interesting. However, some revisions are necessary to meet the requirement for publishing. Detailed comments are listed as follows.
(1) The scientific contribution of this paper is not clear. As far as I know, the combination of DWT and LSTM is not a novel idea in the field of PHM.
(2) It seems that the literature review on anomaly detection, fault diagnosis, and residual life prediction methods based on deep learning is not sufficient. The following papers can be considered.
[1] Lightweight Multiscale Convolutional Networks With Adaptive Pruning for Intelligent Fault Diagnosis of Train Bogie Bearings in Edge Computing Scenarios. 10.1109/TIM.2022.3231325
[2] Online Joint Replacement-Order Optimization Driven By A Nonlinear Ensemble Remaining Useful Life Prediction Method. 10.1016 / j.y MSSP. 2022.109053
(3) In the experiments, the parameters of the intelligent model need to be clearly stated. Further, the authors should explain how they select these parameters.
(4) There are some formatting errors in the paper. For example, the abbreviations after the full names of some nouns lack parentheses. Please check carefully.
Round 2
Reviewer 1 Report
All the best to the authors.